# Micro-Hydropower in Nepal: Analysing the Project Process to Understand Drivers that Strengthen and Weaken Sustainability †

Joe Butchers *, Sam Williamson and Julian Booker 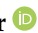

**Abstract:** Evaluating the sustainable operation of community-owned and community-operated renewable energy projects is complex. The development of a project often depends on the actions of diverse stakeholders, including the government, industry and communities. Throughout the project cycle, these interrelated actions impact the sustainability of the project. In this paper, the typical project cycle of a micro-hydropower plant in Nepal is used to demonstrate that key events throughout the project cycle affect a plant's ability to operate sustainably. Through a critical analysis of the available literature, policy and project documentation and interviews with manufacturers, drivers that affect the sustainability of plants are found. Examples include weak specification of civil components during tendering, quality control issues during manufacture, poor quality of construction and trained operators leaving their position. Opportunities to minimise both the occurrence and the severity of threats to sustainability are identified. For the micro-hydropower industry in Nepal, recommendations are made for specific actions by the relevant stakeholders at appropriate moments in the project cycle. More broadly, the findings demonstrate that the complex nature of developing community energy projects requires a holistic consideration of the complete project process.

**Keywords:** stakeholder; community; hydropower; mini-grid; Nepal

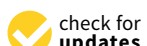



## 1. Introduction

Community-owned renewable energy projects are an option for increasing electricity access in off-grid areas. To deliver electricity services that result in a prolonged impact n lives and livelihoods, schemes must operate sustainably [1]. The literature focused on the assessment of community energy projects has identified that sustainability depends on factors such as technical reliability, financial viability and community engagement [2–4]. Typically, as these studies are conducted at the operational stage, they may not be able to evaluate the emergence of these factors during the project cycle. Elsewhere, research has considered the success of national-level programmes that drive the introduction of renewable energy technologies [5–7]. These two levels, individual project outcomes and the macro-landscape, are connected by the project process. The landscape determines the actions of stakeholders within the project process. For community-owned projects, the process is often long in duration, complex and dependent on multiple stakeholders. In the field of small-scale renewable energy projects, research has considered the institutional landscape and project outcomes; however, they are often evaluated separately. Therefore, there is an opportunity to investigate how the project process (shaped by the institutional landscape) influences the project outcomes.

To explore this, the case study of micro-hydropower development in Nepal is considered. In Nepal, micro-hydropower refers to electricity generation using hydropower at less than 100 kW [8]. A typical micro-hydropower plant (MHP) is shown in Figure 1. There

are approximately 3300 community-owned and community-operated MHPs installed in Nepal [9]. The majority have been funded through subsidies administered by the Alternative Energy Promotion Centre (AEPC); since 2006, the Rural Energy Policy and Subsidy for Renewable Energy has ensured subsidy delivery for renewable energy technologies, including micro-hydropower [10]. From the 1960s, development efforts by international donors have resulted in the creation of a domestic micro-hydropower manufacturing industry, which still produces most of the generating equipment today [11,12]. Schemes are initiated by communities who must contribute financially and physically during construction [13]. Following installation, communities are responsible for owning and operating the plants themselves. Despite the presence of other renewable energy technologies for community electrification (including solar, wind and hybrid systems), in Nepal, micro-hydropower is established as the dominant technology in the hilly and mountainous areas. This paper focuses on micro-hydropower in this context; the selection of other forms of mini-grid technology is not considered.

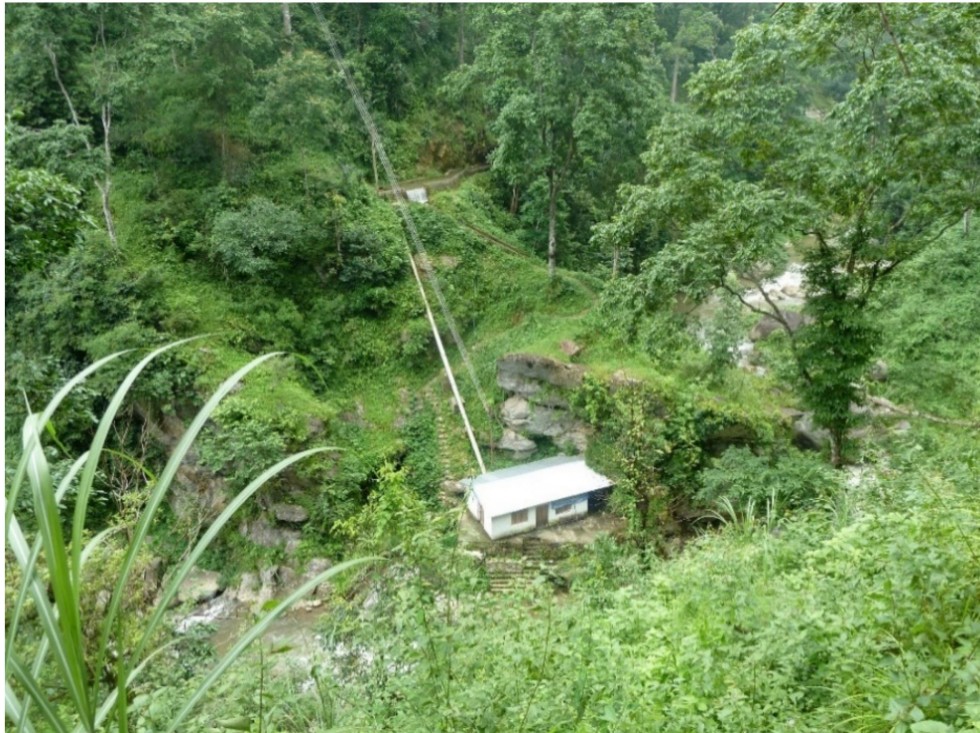

**Figure 1.** A typical micro-hydropower plant. The powerhouse, penstock pipe, civil structures and transmission lines are all visible. Photo credit: Sam Williamson.

The research on micro-hydropower in Nepal has tended to focus on two separate levels: the national landscape and the outcomes of individual projects. At the national level, Nepal's development of micro-hydropower is generally considered a success when compared with other countries. In [5], Nepal's Rural Energy Development Programme—a programme that resulted in the construction of 250 MHPs—is evaluated alongside nine other national renewable energy programmes in the Asia-Pacific region. Of the 10 programmes, Nepal's is amongst 6 considered to be successful. Factors present in Nepal (and common to many of the successful other programmes) include the use of appropriate technology, integration of income generation, availability of subsidies, use of local capacity and the successful engagement of communities. In [14,15], Nepal is mentioned alongside India and Pakistan as having rich experience in micro-hydropower projects, particularly in relation to community involvement. These countries have used similar models of community involvement and ownership to enable project development.

Studies focused on Nepal have analysed the funding mechanism for renewable energy projects, highlighting the success of the subsidy policy in increasing the number of MHP installations [14,15]. However, challenges identified include a cumbersome delivery process for manufacturers and a lack of involvement of the financial sector (largely due to poor loan recovery and shortage of collateral in rural areas). In [16,17], the success of two national-level programmes in Nepal is considered. The promotion of community involvement, the diversity of institutions involved (national and local governments, and community-based) and a focus on maintenance and after-sales are identified as success factors. Whilst focused on small hydropower plants (categorised as between 1 and 25 MW in Nepal), many of the political, economic and technical barriers described recently in [18] are also pertinent at the micro-hydropower scale. Politically, they include the instability of the Nepali government and unclear designation of responsibility for smaller scales of hydropower. Economically, there is a lack of bank financing due to high interest rates and difficulty in securing long-term loans. Technically, projects are often affected by low load factors, whilst high sedimentation rates increase maintenance frequency. These recent findings ([18] was published in 2019) reinforce similar assertions made in 2013 in [19], suggesting that such challenges have maintained a consistent presence and are difficult to overcome.

At the project level, individual assessments of projects have identified the positive effect that MHPs have on rural communities [20,21]. They have been shown to contribute to improvements in income generation [22], reduce reliance on fossil fuels [20] and offer improvements in health and education [21]. Alongside these examples of their impact, field-based studies have also suggested technical, social and economic issues that limit the sustainability of plants. Examples include poor standards of maintenance [23], unequal distribution of project benefits [24] and failure to generate a sufficient income to support running costs [25]. Such issues reduce the ability of MHPs to deliver their intended benefits and may eventually lead to the failure of plants.

At the landscape level, the development of micro-hydropower projects in Nepal is typically seen as a success. The subsidy policy, development of a local manufacturing industry and successful engagement of communities have resulted in the construction of a significant number of projects. However, there is a lack of reliable data to describe the current status of installed projects. Therefore, whilst arguably successful in constructing plants, the efficacy of the project process in delivering plants that are sustainable is not known definitively. Evidence from the field suggests that there are a range of issues that affect plant sustainability, preventing completed projects from maintaining their perceived success in the long term. The objective of this paper is to use evidence from the field to evaluate how the project process influences the sustainability of individual plants. The subsidy-based financing of MHPs in Nepal has resulted in a common project process. Within this, certain elements are unique: the characteristics of the site and the community change from one project to another. However, for every project, the process dictates the actions and responsibilities of various stakeholders. In this paper, the available literature, government documentation and interviews with manufacturing companies are used to understand the roles and responsibilities of stakeholders and the factors affecting sustainability that have been observed at MHPs. By evaluating the stakeholder responsibilities throughout the project cycle, it is possible to understand how these factors develop. This methodology allows the success of the project process to be considered in relation to long-term sustainability rather than project completion. Lessons from this case study can be used to inform other community-owned renewable energy projects, regardless of technology and location.

## 2. Methodology

The objective of this study was to understand the connection between the project process and its outcomes. For project outcomes, the particular focus was the operational sustainability of plants, which here is defined as the ability of the technology and its

stakeholders to deliver electricity services that meet the expectations of consumers over a system's expected life span.

Initially, the methodology used a range of sources to address two key questions:

(1) What is the typical MHP project process in Nepal?
(2) What are the potential project outcomes relating to sustainability?

In relation to the project process, we hoped to understand who the involved stakeholders are, their roles and responsibilities and the key events during the project process. In terms of project outcomes, the intention was to find the range of potential drivers (present at the operational stage) that positively or negatively affect plant sustainability. Subsequently, the findings in these two areas were analysed, allowing connections to be identified. Using these results, it was possible to find opportunities within the project process to tackle the occurrence of negative drivers and reinforce the positive drivers.

The methodology depended on using a diverse range of sources to collect the relevant information. These included primary sources (first-hand experience and interviews) and secondary sources, including government documentation, and academic and grey literature.

### 2.1. First-Hand Experience

In [25], the authors of this article conducted a study to consider factors that affect the sustainable operation of plants at 24 sites. The results of that study, including interviews conducted with plant managers, operators and consumers, have been used in this paper. They provide primary evidence of the roles of stakeholders and outcomes in the field. Observations resulting from [25] that are used in this paper are referenced accordingly.

### 2.2. Interviews with Manufacturers

Semi-structured interviews were conducted with representatives of five micro-hydropower companies. The interviews were conducted with senior employees who were responsible for managing the production of hydro-mechanical equipment. Open questions were intended to explore their actions during the design, manufacture and construction phases, and their response to issues that occur in the field after installation. The interviews were conducted in English and recorded. Ethical assessment was completed prior to the interviews. Interviewees were informed that the information collected was for research purposes only.

### 2.3. Policy and Government Documentation

Table 1 lists the policy documentation and guidelines that are openly available from the AEPC. These documents are broadly of two types: first, those that are lawful; second, those that are supportive to the policy or provide advisory information to stakeholders. Alongside the freely available government documentation, the AEPC and one of the interviewed manufacturing companies provided a total of three tendering documents [26–28]. They describe the details of subsidy eligible projects and provide the specification of sub-systems to be quoted for.

### 2.4. Academic and Grey Literature

In Nepal, both academic research and project reporting by government and non-government organisations has resulted in a large body of information that document specific project outcomes in the field. These sources typically describe the outcomes for a single or multiple projects. All of the sources used considered projects that had been dependent on subsidy-based delivery with active community engagement and eventual community ownership. The sources were used to identify evidence of common stakeholder roles and outcomes that affect the sustainability of MHPs.

### 2.5. Limitations

The available government literature is comprehensive and gives an indication of the expected best practice throughout the project cycle. Without interviewing staff from na-

tional and local governments, it is not possible to evaluate the extent to which government documentation is implemented at the project level. Interviews with representatives of manufacturing companies gave an indication of their perspective. Although reputable and established (each trading for at least 15 years), the sample size of five manufacturing companies represents less than 10% of the companies registered with the Nepal Micro Hydro Development Association [29]. In the methodology, the community perspective has largely been extracted from secondary data. Typically, as project assessments focus on the operational stage, there is a lack of information that describes the views of the community throughout the whole project cycle.

**Table 1.** Policy documentation and guidelines from the Alternative Energy Promotion Centre (AEPC).

| Year | Title | Overview |
| --- | --- | --- |
| 2006 | Rural Energy Policy | Ensures the participation of the local government and creates a rural energy fund for subsidy delivery |
| 2008 | Micro-Mini Hydropower Output and Household Verification Guideline | Advises inspectors on how to verify the power output of micro-hydropower plants (MHPs) at the plant and the household level |
| 2013 | Terms of reference for pre-qualification of consulting companies for survey and design of micro-hydropower projects | Provides the criteria that companies must fulfil to be eligible for subsidy |
| 2013 | Guideline for cooperative model of mini-/micro-hydropower projects | Provides background and instructions for the formation of a mini-/micro-hydropower cooperative |
| 2013 | Micro Hydro Project Construction & Installation Guideline | Provides detailed instructions for construction of civil structures |
| 2014 | Reference Micro Hydropower Standard | Provides the expected standard for hydroelectric-generating sets, associated civil works and electrical transmission and distribution lines with capacities up to 100 kW |
| 2016 | Renewable Energy Subsidy Policy | Provides the subsidy quantities for several renewable energy technologies |
| 2016 | Subsidy Delivery Mechanism Policy | Outlines the process for administering subsidies to renewable energy projects |
| 2018 | Guideline for Detail Feasibility Studies of MHPs | Advises consultants on the standard approach for conducting and reporting on detailed feasibility studies of MHPs |

## 3. Results

The results are presented in three sections. Firstly, the various stakeholders are identified and categorized. Secondly, the key actions for these stakeholders are identified throughout the project timeline. Finally, the positive and negative sustainability drivers are presented.

### 3.1. Stakeholder Roles and Responsibilities

Community energy projects are dependent on multiple stakeholders and their commitment, collaboration and alignment [30,31]. During the project cycle, stakeholders' actions and their perception of the project influence both the process and the outcome [32]. The experience of the authors and evidence in [13] allow the stakeholders to be categorised into three groups:

**Institutional:** Institutionally, there are multiple stakeholders acting at the national and local levels. Nationally, the AEPC is the government agency that supports renewable energy technology in Nepal. It administers subsidies and provides technical support to

individual communities and regional government offices. Working alongside the AEPC, the Nepal Micro Hydro Development Association represents approximately 60 of the micro-hydropower companies based in Nepal [29]. It advocates for the interests of these companies and regulates the training that is delivered to plant operators and managers. At the local level, district coordination committees (DCCs) are government bodies that represent the interest of local communities within a single district [13]. They usually provide financial support to renewable energy projects that occur within their district. Specifically working to improve access to renewable energy technologies, regional service centres (RSCs) provide an on-the-ground presence to advise and support communities. There are 10 RSCs that cover the 77 districts of Nepal [33].

**Community:** The community is comprised of people from a single or multiple villages who are interested in developing an MHP together. The interests of the wider community are represented formally through a micro-hydro functional group or a cooperative (MHFG/C). In the cooperative structure, financial contribution by the member gives them a share in the MHP, whilst in a functional group, it is not formally acknowledged. Within this study, the differences between the ownership models are not considered in detail. From the community, several plant operators and a plant manager are chosen to be responsible for the operation and maintenance (O&M) of the MHP once the installation is complete. It should be noted that members of the community are heterogenous (particularly in status and wealth) [24,34,35], which affects their perception of, and the actions that they take in relation to, the MHP. In addition, there are existing social structures and local dynamics that affect the process of MHP development and its outcomes.

**Industry:** Within industry, there are technical and financial stakeholders. Consulting companies (CCs) are responsible for conducting feasibility studies, sizing the overall scheme, specifying key components and designing the civil structures [13]. Manufacturing and installer companies (M/ICs) produce or procure the required hydro-mechanical and electrical equipment. In Nepal, it is common for companies to perform all three of the technical services of consultation, manufacturing and installation. Private finance institutions and banks provide credit to local communities to pay for project costs that are not covered by the subsidy [36].

A stakeholder onion diagram can be used to represent the position of stakeholders in relation to a particular goal [37]. Figure 2 shows the stakeholder groups and the individual stakeholders within this. Table 2 identifies the stakeholders, their groups and acronyms. The colours of stakeholder groups in the table correspond to the diagram. At the centre of the diagram is the goal that all the stakeholders are working towards. In this context, the goal is the installation and operation of a sustainable MHP. The first level outside of the MHP is the community: the stakeholder group that most directly interacts with the goal. The next level is shared between local institutional and industrial stakeholders. These stakeholders design, develop and facilitate the installation and integration of the MHP within the community; they also continue to have some involvement after the installation is complete. The outer level includes national institutional stakeholders who administer financial and technical support. For the purposes of this study, the boundary is drawn at this level. However, it should be considered that the Ministry of Energy, Water Resources and Irrigation and international donors have a significant influence over the AEPC's direction and approach [33].

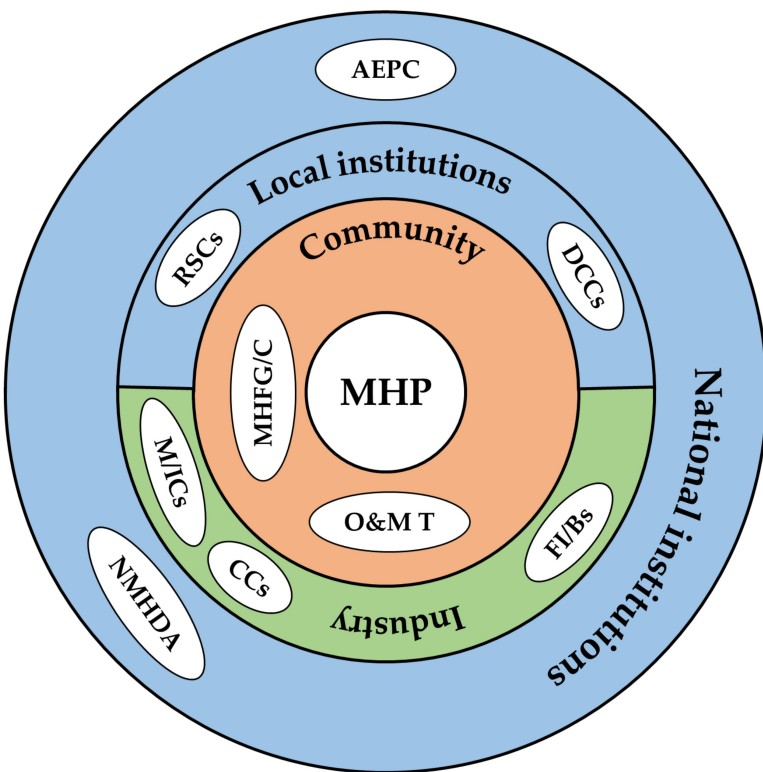

**Figure 2.** Relationships between stakeholders for the installation and operation of an MHP.

**Table 2.** Identification and categorisation of stakeholders. The colours shown are used to identify the 3 stakeholder groups.

| Group | Stakeholder | Acronym |
|---|---|---|
| Institutional | Alternative Energy Promotion Centre | AEPC |
| | Nepal Micro Hydro Development Association | NMHDA |
| | Regional service centres | RSCs |
| | District coordination committees | DCCs |
| Industry | Manufacturing and installation companies | M/ICs |
| | Consulting companies | CCs |
| | Financial institutions and banks | FI/Bs |
| Community | Micro-hydro functional g/cooperative | MHFG/C |
| | Operation and maintenance team | O&M T |

### 3.2. Project Timeline

The subsidy amount available to the community is determined based on the number of households to be electrified and the overall rated power of the scheme [38]. The districts of Nepal are placed into four categories based on their remoteness, with the subsidy amount varying accordingly [38]. Typically, the subsidy covers around 50% of the total project cost; the remaining balance is usually comprised of the community's labour and financial contribution, donations from local governments and bank loans [13,33,36].

Using the information from the subsidy documentation and supporting literature, Table 3 shows the key actions required by the stakeholder groups throughout the project process. The actions listed are given in approximately sequential order but may occur concurrently. The project process is considered in five distinct phases: project initiation, design and manufacture, construction, installation and commissioning, and operation. The milestones prescribed by the *Subsidy Delivery Mechanism Policy* are indicated in the table (highlighted in grey), with the specific action required by the M/IC shown in italics.

**Table 3.** Actions and responsibilities of stakeholder groups throughout the project process.

| | Institutional | Industry | Community |
|---|---|---|---|
| **Project initiation** | | | Community makes an application to RSC or AEPC directly |
| | RSC carries out pre-feasibility study | | Community registers MHFG/C |
| | RSC recommends to AEPC that a detailed feasibility study (DFS) take place | | |
| | RSC assists in selection of pre-qualified CC | | MHFG/C selects pre-qualified company to conduct DFS |
| | | CC conducts DFS and submits report to RSC | MHFG/C submit business plan for the MHP |
| | RSC and AEPC decide to accept DFS business plan and approve subsidy | CC receives payment for DFS from AEPC | |
| | | | MHFG/C begins to collect funds and deposits in a community account |
| | RSC calls for bids from pre-qualified companies | M/ICs submit bids based on tender documentation | |
| | | | MHFG/C selects M/IC |
| | *Milestone*: payment of 30% instalment | M/ICs submit bank guarantee | |
| | | | Selection of operators and managers |
| **Design & manufacture** | | Design by M/ICs | |
| | | Manufacture of electro-mechanical equipment by M/ICs | |
| | *Milestone*: payment of 45% instalment | M/ICs deliver equipment to site | |
| **Construction** | RSC support civil works and may report to AEPC | M/ICs supervise civil works | Civil works by MHFG supervised by M/ICs |
| **Installation & commissioning** | | Installation by M/ICs | |
| | Power output verified by RSC | Power output testing by M/ICs | |
| | *Milestone*: payment of 15% instalment | M/ICs submit power output report | |
| | Power output verification conducted by a 3rd party | | |
| | NMHDA/CC train operator | | Operator receives training |
| | NMHDA/CC training manager | | Manager receives training |
| **Operation & maintenance** | | M/ICs provide assistance in repair and maintenance | Operation and maintenance of system |
| | *Milestone*: payment of 10% instalment—Final test of power output after one year | M/ICs submit one-year warranty report | |

Rows highlighted in grey are used to indicate the timing of milestone payments and associated activities.

### 3.3. Development of Sustainability Drivers

Using the available sources, it was possible to identify outcomes observed in the field that influenced the sustainability of MHPs. These could be categorised as positive drivers—factors that improve MHP sustainability—and negative drivers—factors that reduce MHP sustainability.

Table 4 lists the sustainability drivers that were identified from the available sources. Within each list, some of the identified drivers are directly contradictory. As the evidence for these drivers came from a range of sources, contradictory drivers can occur at separate plants. Furthermore, given the dynamic nature of both the technology and the socio-economic landscape it resides in, similar drivers could occur at the same MHP but at different times. Elsewhere, relationships may exist between the identified drivers, e.g., *Insufficient income to pay for repairs* is connected to *Beneficiaries not paying regularly*. However, as each driver may develop for a range of reasons and have multiple causal effects, all are deemed worthy of consideration. The drivers are derived from empirical field-based evidence, as there are little numerical data regarding the status of installed MHPs in Nepal.

For most drivers, multiple sources are presented as evidence. These individual sources often provide information from multiple MHPs. Consequently, the identified drivers are not unique outcomes (specific to an individual plant); rather, they are considered feasible outcomes at any installed MHP in Nepal.

**Table 4.** Drivers affecting sustainability identified at the operational phase.

|  | Observation | Evidence |
|---|---|---|
| Negative drivers | Civil structures require repair due to landslides and monsoon | [25,34,39] |
|  | Poor standard of civil construction | [13,25,39] |
|  | Misalignment of rotating components | [25] |
|  | Poor standard of maintenance | [23,25] |
|  | Insufficient income to pay for repairs | [25,40] |
|  | Uneven distribution of benefits | [24,34,35] |
|  | Conflict within the community—water/land/political | [24,25] |
|  | Community not supportive in repair work | [23,25] |
|  | Reduced power output | [41] |
|  | Low load factor | [13,22,39,40] |
|  | Problems with tariff collection | [25,34,39,40] |
|  | Beneficiaries not paying regularly | [23,25] |
|  | Untrained operator | [25,42] |
|  | Alternative energy sources are available | [13,25,34] |
|  | Poor functioning of MHFG | [39] |
|  | Insufficient flow rate | [25,39] |
|  | Misuse by consumers | [23,25] |
|  | Hydro-mechanical equipment failure | [23,39] |
|  | Low tariff setting | [13,25,34] |
|  | Distance to repair centres | [23,40] |
|  | Lack of proper accounting | [22,40] |
| Positive drivers | Effective collection of tariffs | [25,34] |
|  | Consumers pay regularly | [25,40] |
|  | Plants deliver benefits to community | [20,25,35,39,42] |
|  | Use of electricity meters | [25,39] |
|  | Good sense of ownership amongst community | [25,34,42] |
|  | Trained operator | [13,25,39] |
|  | Trained plant manager | [40] |
|  | Installed equipment delivers expected rate of power | [25,41] |
|  | Supportive community attitude | [25,34] |
|  | Good relationship with M/ICs | Interviews with M/ICs |
|  | Plant funds are correctly managed | [25,40] |
|  | MHFG is institutionally strong | [16,39,42,43] |
|  | Community willing to assist with repairs | [25,40] |
|  | High load factor | [40] |
|  | Range of productive end uses | [25,40] |

## 4. Discussion

To consider the occurrence of the drivers in Table 4, the project process and stakeholder actions are discussed in relation to the following areas: responsibilities, capacity, quality control and the local environment. Whilst initially discussed in separate sections, the recommendations address some of the overlap and interactions between these areas.

### 4.1. Responsibilities

Throughout the project process, various stakeholders have responsibilities to fulfil. Prior to commissioning, there is significant interdependence between the stakeholders' responsibilities; specific actions are contingent on one another. Following commissioning, most responsibilities lie with the community, with only occasional support from the M/IC when technical problems occur. During implementation, the responsibilities are usually clearly defined due to the milestones imposed by the *subsidy delivery mechanism* [38].

When the responsibility is not clearly defined, it can be problematic. For example, the construction of civil structures from intake to forebay tank is considered to be primarily the responsibility of the community [13]. However, within tendering documentation, supervision of all civil works is an item line that M/ICs must quote for [26–28]. Alongside this, RSC engineers may also be expected to support the installation [13], leading to a lack

of clarity in accountability and resulting in higher potential for poorly constructed civil structures. This stage of construction highlights a broader problem: it is difficult for the community (despite being project owners) to hold other stakeholders accountable.

Many of the responsibilities in the early phases of the project process result in physical outputs that are checked by institutional stakeholders. Alongside this, the actions of the community contribute to a less tangible but vital outcome: the development of collective responsibility for the MHP. Without this, weaknesses like internal conflict, lack of support in repair and irregular payment are likely to arise. Throughout the project process, certain actions are supportive to fostering the engagement of the community. At the outset, the formation of an MHFG/C aligns the interest of the community, provides representation to marginalised groups and creates a platform for community interaction with the other stakeholders [29]. The MHFG/C should ensure that all beneficiaries are active during the project, but it is also its responsibility to continue to engage the community after installation. Failure to arrange public meetings and engage beneficiaries leads to a loss of interest [44].

Monetary investment is useful in engaging individuals, and as this is expected (at an appropriate level) from all beneficiaries, it is an opportunity for all households to contribute [13]. The community responsibility of the civil construction reinforces individual commitment to the collective cause. At this stage, physical rather than monetary commitment is required, with some community members working for at least 6 months. These actions are important in developing a collective responsibility for the plant. The members of the community selected to be managers and operators have a greater responsibility. Technically, if plant operators fail to conduct regular maintenance, the reliability of the plant will suffer [25]. Economically, plant managers must ensure that tariffs are collected regularly and the plant's income is managed. Without these actions, negative drivers can develop, and the sustainability of the plant is likely to suffer. Whilst operators and managers are paid for their work [45], a large amount of responsibility is attributed to these individuals.

### 4.2. Capacity

The drivers at the operational stage are often related to the stakeholders' capacity to perform their responsibilities. The institutional framework and some stakeholders (e.g., the AEPC) remain constant from one project to another. For most of the other stakeholders, their capacity is variable. Amongst both M/ICs and RSCs, there is variation in competence, experience and manpower. At the outset, the community possess a certain capacity (e.g., financial status, cohesion, presence of managerially and technically experienced people), but the project process is likely to alter this.

From the community, several people are chosen to receive training for the roles of operator and manager. Their selection by the MHFG/C affects the reliability and financial sustainability of the plant at the operational phase. In some cases, plant operators are selected for social and economic reasons. For example, their land might be in use for the powerhouse, or they are related to someone in a position of authority [34]. In these cases, they may not possess the motivation or capacity of someone chosen through a fair selection process. Training of managers and operators is required to ensure that they are competent to fulfil their roles. For operators, there is a 22-day course [46], which teaches them how the system operates, and regular preventative and corrective maintenance procedures. This has been common for over 20 years [46] and has been shown to have a positive impact on the reliability of operational schemes [45]. However, it is common for men to move away to find employment, and if a trained operator leaves, the knowledge acquired during training (and informally during the construction and installation phases) is lost [23,45]. Evidence in the literature suggests that the training of plant managers is not as regularly practiced as operator training. For example, in [40] only 43% of managers had been trained compared to 100% of operators. This is likely to contribute to a range of the observed weaknesses, such as problems with tariff collection, low tariff setting and lack of proper accounting.

During construction, it is the community's responsibility to collect raw material and build the civil works. The interviewed manufacturers explained that poor-quality materials are often collected and that a lack of trained skilled labour affects the precision that civil structures are built to. This results in weaknesses in both the quality of the civil structures and their ability to perform certain functions, e.g., extraction of silt in the de-silting bay [13]. The construction of the civil structures by the community is intended to reduce the overall project cost, with only supervision provided by M/ICs [26]. However, according to one interviewed M/IC, as the level of supervision is not dictated, the technicians sent to site often lack the knowledge and experience of civil elements. Alongside the M/ICs, RSCs are expected to provide ongoing support and ensure that the construction is taking place as planned. Often, RSCs do not have enough staff with the relevant experience to provide a consistent presence on-site [13]. Developing an understanding of the quantity and capacity of RSC staff would be useful in determining their ability to provide the requisite level of support.

The actions of M/ICs are largely prescribed by the subsidy process; interviewed manufacturers explained that they do what is required to receive the subsidy. New companies have entered the market, but they focus on cost reduction rather than innovation [13]. The result is that M/ICs continue to produce similar designs with the same equipment, without looking for opportunities to introduce new manufacturing processes [39] or bought-out components.

### 4.3. Quality Control

Quality control (QC) processes are important in ensuring that actions have been completed to a required standard. In this context, the term quality control is used to describe "any process for maintaining a desired quality of product or output" [47]. Such processes ensure that for individual projects, quality issues can be identified and that from one project to another, there is replicability. As the project funder, it is the responsibility of the AEPC to implement QC processes. Manufacturers may conduct some internal QC processes, but their actions are mostly dictated by the subsidy policy. The AEPC has produced an extensive range of guidelines that describe its expectations for how multiple phases of the project process should be completed [48–50]. These are comprehensive examples of good practice that when followed can motivate the creation of positive drivers. Alongside the guidelines, there are multiple QC processes, including several that are directly related to the delivery of subsidies. As the government administers both the documentation and the quality assurance, there needs to be correlation between these two areas.

Outside of the project cycle, the AEPC pre-qualifies both and M/ICs [13,51]. Pre-qualification is used to assess whether companies possess the human resources and experience required. From the detailed feasibility study (DFS) stage, the guidelines demonstrate what should be included in the report [50]. Following the submission of this report, a technical review committee (TRC) comprising interdisciplinary stakeholders assesses the report, providing an early opportunity to flag technical, social and economic issues [38].

A tendering document provides specification of all the sub-systems of an MHP. In the case of some sub-systems, such as the turbine and generator, the specification is clearly defined and can be checked [26]. For the civil structures, whilst drawings are provided, the available manpower at RSCs is a barrier to regularly checking their construction [13]. Whilst the on-paper design for the civil structures is checked by the TRC, the timing of the final check after installation means that if there is an issue, remediation may be expensive and time consuming. The final subsidy payment depends on measuring the output performance of the MHP and a visual check of the quality of the installation [52]. Often, the measurement on-site results in a value for the overall output power and not the hydro-mechanical efficiency. As such, it is difficult to compare the equipment of different manufacturers. Furthermore, the inspection of equipment only occurs on-site after it has been installed. A manufacturing or assembly defect that is observed at this stage cannot

be rectified. There are standards for the manufactured equipment [8], but these are not referred to within subsidy documentation [38,53], and it was not possible to find evidence of its use for assessment elsewhere within the literature.

During the project process, there are multiple activities that consider the financial viability of the project. Initially, the submittal of a project business plan ensures that the MHFG/C considers the importance of the plant's economic operation. In the DFS, the CC quantifies the consumer's willingness to pay and the opportunities for productive end uses in the local area [50]. Observation of the business plan and assessment of the DFS ensure that institutional stakeholders have considered the financial viability alongside the technical feasibility. However, between the TRC review and training of the plant manager, there are no activities that consider whether the business plan has been implemented.

### 4.4. Local Environment

The project process dictates the roles and responsibilities of the stakeholders, but many outcomes are also related to the physical and socio-economic landscape that a project develops in. The physical landscape affects the rated power available, the location and form of the sub-systems and the proximity of the plant to the beneficiaries. The socio-economic landscape dictates factors such as the wealth of beneficiaries, the opportunities for productive end uses and existing cohesion within the community. All of these factors influence both the project process and its outcomes.

At the operational phase, the site's geographical features affect the seasonal water flow and the frequency of landslides and flooding. The DFS considers the geography [50], using an appropriate design as mitigation (e.g., storm traps to mitigate the effect of landslides), but some sites remain at greater risk or require more regular maintenance. The location of the site in relation to the community is also significant. At some MHPs, beneficiaries can be located at a 6 h walk from the powerhouse [45]. During the construction, it may be difficult to mobilise community members who are physically far away. At the operational stage, it may impact the jobs of operators and managers and the willingness of community members to pay or participate in meetings and repair works.

The socio-economic status of the local community is also relevant. In larger settlements, it is easier to connect a greater range and number of productive end uses [45], increasing the plant's load factor and its income. Proximity to the beneficiaries may also affect tariff collection. If beneficiaries can pay at a location near to their home, they are likely to pay more regularly. A potential negative impact is that in larger settlements, a higher proportion of people depend on businesses rather than farming for their livelihoods [23]. They may be more resistant to supplying labour during the construction and for repairs when required [23]. Furthermore, in larger settlements, it may be more difficult to mobilise the community collectively. Some MHPs have very scattered beneficiaries. Often, communities located away from roads are likely to be of lower socio-economic status. They may struggle to contribute financially, with both initial and then recurring payments. This can be compounded by the distributed location of the households, which increases the difficulty in collecting tariffs.

### 4.5. Recommendations and Lessons Learned

In Nepal, the micro-hydropower project process demands active participation and collaboration from multiple stakeholders. The subsidy-driven process has led the AEPC to develop documentation that details standards and quality assurance, but the capacity of the institutional stakeholders is a barrier to implementing them rigorously. As a result, the quality of key technical components is often not checked until after they have been installed. The M/ICs interviewed during this study possess the experience and capacity to deliver reliable technical systems. They are capable of manufacturing equipment to the standards set by the AEPC and supervising the community in the construction of civil works. However, the current subsidy structure means many projects are given to the lowest bidder, which drives down the quality of technical elements. The community actions are

effective in fostering engagement and result in the completion of MHP construction, but supporting actions are required from other stakeholders to ensure that the actions of the community result in sustainable projects. Currently, the creation of productive end uses and the financial management of plants are a particular weakness observed widely in the literature. Between different sites, the potential for productive end uses is highly variable and can be identified early in the project cycle. Based on the findings of this work, the following recommendations are made, which are considered feasible within the current project structure:

- Training of plant managers is essential and should be practiced at every new installation. It should be conducted locally by RSCs to maximise the number of participants.
- On behalf of the AEPC, independent consultants should use the Reference Micro Hydro Power Standard to check the adherence, quality and key dimensions of manufactured and bought-in hydro-mechanical equipment before it is dispatched to site.
- Civil structures should be formally checked against the project drawings and AEPC standards by the RSC during construction and before commissioning. A subsidy payment to the M/IC for the supervision of civil works should depend upon it.
- The business plan should include clearly defined actions that can be checked by the RSC. Sites with low potential for economic activity should be identified and supported. A second-stage business plan that indicates progress should be submitted when the equipment is delivered to site.

In general, for other community energy renewable energy technologies, the established project cycle in Nepal is able to provide a number of lessons. The initiation of the project by the community and its ongoing involvement are effective in fostering ownership. Finding a financial or physical contribution that is appropriate for each household is important. A subsidy-driven process provides an opportunity to introduce quality control mechanisms. However, administering these effectively requires sufficient capacity and is more effective if administered at the local level. Each project develops within a socio-economic and physical landscape that affects the project process and its outcomes. To operate sustainably, the location of some schemes means that they require greater support during the project process. Proper evaluation of the market opportunities and ongoing support to introduce productive end uses are important in ensuring that plants have high load factors and generate sufficient income. Furthermore, the responsibility of operation and maintenance usually resides with a handful of individuals; they must be properly trained and fairly paid.

In many contexts in the Global South, accurate information describing the status of installed energy projects is difficult to access. The methodology employed in this paper has sought to use field-based evidence to evaluate how project outcomes are determined by the project process. An opportunity for further work is to apply this methodology in different contexts. In countries where a national programme is used to deliver energy projects, the methodology could be applied to understand the efficacy of the project processes in delivering sustainable energy access, regardless of location or technology.

## 5. Conclusions

Comprehensively understanding the development of sustainability drivers requires evaluation of the institutional landscape, project process and stakeholder roles. In this paper, the case study of micro-hydropower plants in Nepal has been used to show that operational drivers can be connected to events that occur within the project cycle. The responsibilities of stakeholders, their capacity to fulfil them and quality control processes were identified as key factors in determining the development of sustainability drivers. For Nepal, recommendations include integrating actions that develop financial viability earlier in the project process, ensuring that quality control processes happen at the correct time and ensuring that plant managers are correctly trained. Further work will involve conducting a detailed survey of the capability of manufacturing companies to understand the development of hydro-mechanical defects and to look for opportunities to improve reliability. For community-owned energy projects elsewhere, this works demonstrates the

importance of understanding the influence that the project development process and the interaction of stakeholder responsibilities have upon project outcomes.

**Author Contributions:** Conceptualization: J.B. (Joe Butchers); methodology: J.B. (Joe Butchers); formal analysis: J.B. (Joe Butchers); investigation: J.B. (Joe Butchers); writing—original draft preparation: J.B. (Joe Butchers); writing—review and editing: S.W. and J.B. (Julian Booker); supervision: S.W. and J.B. (Julian Booker); project administration: J.B. (Joe Butchers); and funding acquisition: S.W. and J.B. (Joe Butchers). All authors have read and agreed to the published version of the manuscript.

**Funding:** This study was funded by the Cabot Institute for the Environment and a studentship from the Engineering and Physical Sciences Research Council (award reference 188052).

**Institutional Review Board Statement:** The study was conducted according to the guidelines of the Declaration of Helsinki, and approved by the Faculty of Engineering Research Ethics Committee of University of Bristol (reference 83144, 7 March 2019).

**Informed Consent Statement:** Informed consent was obtained from all subjects involved in the study.

**Data Availability Statement:** All underlying data are provided in full within this paper.

**Acknowledgments:** The authors would like to acknowledge the People, Energy and Environment Development Association for its assistance in conducting interviews and all of the interview participants for their time and contributions.

**Conflicts of Interest:** The authors declare no conflict of interest. The funders had no role in the design of the study; in the collection, analyses or interpretation of data; in the writing of the manuscript; or in the decision to publish the results.

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
