# Peer review of "Micro-Hydropower in Nepal: Analysing the Project Process to Understand Drivers that Strengthen and Weaken Sustainabilityâ€"

_sustainability, doi:10.3390/su13031582_

Round 1
Reviewer 1 Report
1. For the first look, the aim of this paper is interesting and the author tried to give a broad idea of MHP and community development in Nepal. It will be helpful to other developers to get an idea of Nepal MHP. However, the author could not follow the general information of MHP and SHP(Small Hydropower) development in Nepal. Please describe the current condition, policy analysis (especially national level) and barriers to MHP and SHP development in Nepal (see: World Small Hydropower Development Report 2019 which is published by UNIDO and ICSHP). 2. Since the main topic is "Development of strength and weakness", this paper did not much explain about strength and weakness. There is a list of strength and weakness in the table-4, but the â‘ many points are contradictory, â‘¡ the author did not explain, what were the criteria to list something as a strength and something as the weakness? I think it would be better if the author explains operational strengths and weaknesses on each level (e.g. national / local policy level, community level, design level, technology level, Quality & management level). Also, there must be clear evidence of some Major strengths and major weaknesses with reference to similar models running in other countries. Please discuss with the details of strengths and weaknesses. It is too short to persuade. 3. In table-3, the Formation of MHFG/C is not clear. Is it formed after PFS by AEPC? It has been written that " MHFG/C selects prequalified company to conduct DFS", Who pays for this? is it bear by communities or done by government support? I could not understand this table clearly. 4. Topic number 3.2 repeated two times, please change it (3.2 → 3.3).Author Response
Please see the attachment.

Reviewer 2 Report
The paper sets out to study barriers and bridges to well-functioning, community-owned micro hydropower installations in Nepal. You claim that this requires a holistic approach. I agree, but I think you miss the first, critical step. Hydropower is just one of the possible options for a community-owned energy project. Thus, there needs to be a screening process before one can proceed with a hydropower project. It is rather obvious that hydropower will not be the best solution at every place, or for every need for energy. You do present some cases where this process seems to have been inadequately performed.
You claim that most of the literature has neglected how the institutional landscape tends to shape the responsibilities of individual stakeholders. Therefore, you will pay attention to these issues, and also assess the capabilities of the stakeholders to shoulder their respective responsibilities.
In Nepal, most micro hydropower projects have been funded through subsidies administered by the Alternative Energy Promotion Centre (AEPC), and since 2006, subsidies for micro hydropower has also been available from the Rural Energy Policy and Subsidy for Renewable Energy program that has ensured subsidy delivery for renewable energy technologies including micro-hydropower. From the 1960s, international development aid has helped to develop an in-country micro-hydropower manufacturing industry, which still produces most of the generating equipment today. The current subsidies stipulate that following installation, communities are responsible for owning and operating the plants themselves.
As you mention, many of the operational problems have their origin in the construction phase. It seems to me that the project cycle is well designed on paper but, still, things can go wrong due to poor monitoring and poor craftmanship. You provide a flora of anecdotal evidence of what can go wrong, but you forget to mention what can help the process to run smoothly. The trainers and facilitators are not accountable to the project owners, but to their respective employers. That’s a surefire way to invite actors to cut corners.
As I understand it, the funding agency has inspectors that are supposed to oversee the project development. A key question is if there is enough of them, and if they have the needed qualifications to assess whether a project is developing properly or not.
At the community level, it is vital to ensure that the community has a satisfactory capability to ensure an economic and technical capacity to manage both the operation and the maintenance of the total system.
My recommendation is that you make a more balanced analysis of the strengths and weaknesses of micro hydropower empoyment in Nepal.
In addition, I have some particular qustions:
- You claim that you studied the “available literature”. What do you mean by that?
- Where did you get you recommendations from (line 384-395)?
- What is the contribution of this paper beyond what is presented in ref 29? Why didn’t you use some real life observations from that paper to support your claims in this paper?
Round 2
Reviewer 1 Report
- Title must be changed. Because, this paper did not discuss the Mini-grid. This paper only focus on Micro-hydropower.
- The idea of "driver" is nice to understand the operational sustainability of MHP. Table 4 and followed discussion point out the positive and negative drivers. It is interesting. But, there is a question about reliability of this analysis, because many elements of table 4 relies on mainly one paper (Reference No. 28). Reconsider the evidence on other arguable grounds.
- Papers of No.28 and 29 are duplicated in Reference section.
